# Lipidomic and in-gel analysis of maleic acid co-polymer nanodiscs reveals differences in composition of solubilized membranes

Marta Barniol-Xicota [1] & Steven H. L. Verhelst [1,2✉]

Membrane proteins are key in a large number of physiological and pathological processes. Their study often involves a prior detergent solubilization step, which strips away the membrane and can jeopardize membrane protein integrity. A recent alternative to detergents encompasses maleic acid based copolymers (xMAs), which disrupt the lipid bilayer and form lipid protein nanodiscs (xMALPs) soluble in aqueous buffer. Although xMALPs are often referred to as native nanodiscs, little is known about the resemblance of their lipid and protein content to the native bilayer. Here we have analyzed prokaryotic and eukaryotic xMALPs using lipidomics and in-gel analysis. Our results show that the xMALPs content varies with the chemical properties of the used xMA.

[1] Laboratory of Chemical Biology, Department of Cellular and Molecular Medicine, KU Leuven, Leuven, Belgium. [2] Leibniz Institute for Analytical Sciences ISAS, e.V., Dortmund, Germany. ✉email: steven.verhelst@kuleuven.be

Membrane proteins (MPs) are essential players in a wide variety of essential physiological processes. Certain specific MPs have been linked to various human diseases. For example, the membrane-bound kinase epidermal growth factor receptor plays a role in different types of cancer and the malfunctioning of the ion channel N-Methyl-D-Aspartate receptor contributes to Alzheimer's disease pathophysiology. Overall, MPs form two-thirds of the total druggable targets in the cell[1,2]. Unfortunately, MPs are less well understood than their soluble counterparts because of their more difficult expression and purification procedures[3].

Purification of MPs requires a prior solubilization step. Traditionally, this has been achieved with detergents, which isolate MPs in micelles by destroying the lipid bilayer. This de-lipidation can compromise structural integrity and may cause problems with protein activity or stability.

A decade ago, styrene maleic acid (SMA) copolymers were reported as nanodisc-forming agents. SMAs can extract MPs with their lipid microenvironment, forming styrene maleic acid lipid particles (SMALPs) soluble in aqueous buffer. Hence, SMAs can fully by-pass the use of detergents. In addition to SMA, several other maleic acid copolymers (xMAs) have been synthesized as nanodisc-forming agents (Fig. 1). These new xMAs have improved chemical properties or incorporate chemical modifications that broaden the downstream applications of xMA lipid protein nanodiscs (xMALPs). For example, diisobutylene maleic acid (DIBMA) has aliphatic alkyl chains instead of aromatic groups, avoiding ultraviolet (UV) absorption and allowing the use of optical spectroscopic techniques such as circular dichroism. Another example is SMA-SH[4], the structure of which contains thiol groups that can act as chemical handles for reaction with fluorophores or other functional moieties.

Since their introduction, xMALPs have resulted in various exciting findings. For example, they enabled the structure determination of the alternative complex III by cryogenic electron microscopy[5] and the elucidation of naturally occurring oligomeric states of various MPs[6].

In our previous work, we found xMALPs to stabilize fragile rhomboid proteases that self-process when in detergent micelles[7]. In addition, rhomboid activity resembled more that in the membrane when in xMALPs than when in micelles. However, we observed that the activity differed depending on the chemical properties of the xMA used. This finding made us question to what extent xMA nanodiscs resemble the native membrane and what lipids are present in the xMALPs. Only a few studies have analyzed the lipid content of SMALPs, focusing on phospholipids (PLs). Hence, the available information on the lipid content of xMALPs derived from biological membranes is very limited, and it is unclear whether the different xMAs show preferential solubilization of certain protein species or lipid classes.

Here we made a detailed comparison of chemically diverse xMA polymers in their solubilization properties of prokaryotic and eukaryotic membranes. In particular, we quantitatively and qualitatively compared the protein solubilization and performed mass spectrometry (MS)-based lipidomics to identify the lipid composition. We found that xMAs display preferential solubilization of proteins and lipids depending on their chemical structure. This study has broad implications for future study of MPs in xMALPs, may help selecting xMAs with desired properties and aid future development of new xMAs with improved characteristics.

## Results

A range of xMAs are currently available as nanodisc-forming agents, with SMA as the most often used polymer[8]. To date, unmodified SMA and DIBMA are the only agents proven to directly extract and solubilize proteins from their native membrane. Here, in order to compare polymers, we selected SMA(3:1), SMA (2.3:1), SMA-QA and DIBMA (Fig. 1), as these represent different chemical features. Whereas DIBMA is completely aliphatic, SMA contain aromatic groups derived from the styrene monomers that show strong UV absorption. In order to compare the effect of the styrene versus maleic acid ratio, we chose for SMAs with ratios 3:1 and 2.3:1. Lastly, in order to assess the effect of charge, we included SMA-QA, which bears a positively charged quaternary ammonium ion instead of the two negative charges from the maleic acid moieties. This polymer was synthesized in-house from a 10,000 g/mol SMA(3:1) anhydride (see "Methods" section for details). The polymers used for solubilization display different compatibilities with buffering systems. For example, SMA-QA is compatible with low pH[9], whereas solubilization by DIBMA is stimulated by higher pH[10]. To prevent any favorable or unfavorable effects of the buffer composition, we chose for buffer conditions that are tolerated by all polymers. Specifically, these comprised 50 mM HEPES pH 7.8 and 300 mM NaCl. We chose for a 2% (w/v) total xMA concentration during solubilization, as this concentration has shown efficient solubilization using different xMAs. In this study, we solubilized prokaryotic and eukaryotic membranes using the four above-mentioned xMAs as well as n-dodecyl-β-D-maltoside (DDM), a mild, non-ionic detergent that is often applied in solubilization of MPs. Specifically, we used membrane pellets of Escherichia coli, one of the most utilized species for recombinant protein expression, and of Jurkat cells, representing a cell line from human origin. In order to eliminate the influence on our experiments from culturing and lysing cells in different batches, we prepared one large batch of membrane pellets from each cell type, which was subsequently used in all solubilization studies.

**xMAs are efficient solubilizing agents for mammalian and bacterial membranes.** Since the first report that SMA copolymers solubilize MPs[11], their use has tremendously grown[12] because of the attractive property of preserving annular lipids around the solubilized MP. Obviously, the efficiency of a solubilization method is an important aspect to consider. We therefore compared the protein solubilization efficiencies of the selected copolymers and DDM, for prokaryotic and eukaryotic membranes, by measurement of the protein concentration using a standard colorimetric assay. For E. coli membranes, DDM is the most efficient method with 70% of total protein being solubilized, and although there is a significantly lower efficiency for the xMA polymers, all but SMA-QA still solubilize >50% of total MPs (Fig. 2a). This trend changed in case of the solubilization of Jurkat MPs, where the xMAs did not have significantly lower efficiencies than DDM. The only exception was SMA-QA, which was considerably less

**Fig. 1 Chemical formulas of the nanodisc-forming polymers used in this work.** Unmodified SMA: $m = 2.3$ or 3, $n = 1$. DIBMA: $m = 1$, $n = 1$. SMA-QA: $m = 3$, $n = 1$.

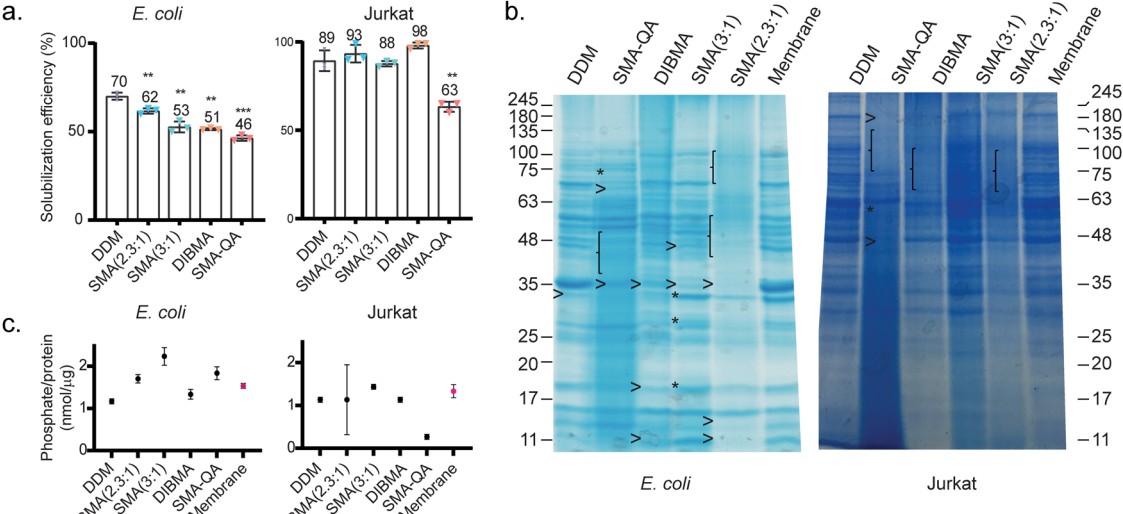

**Fig. 2 Solubilization of *E. coli* and Jurkat membrane pellets with DDM and different xMAs. a** Protein solubilization efficiency, as measured by BCA assay of soluble and insoluble fractions after solubilization with DDM or xMAs. Data correspond to three technical replicates. Error bars represent ±S.D. Significant differences (Student's *t* test) are denoted as **$p < 0.01$, or ***$p < 0.001$. **b** Coomassie-stained gels of membrane protein samples solubilized by different xMAs or DDM. An arrowhead or accolade indicates proteins that are substantially lower abundant than in the membrane pellet. Stars indicate examples of protein bands solubilized by one xMA but not or much worse by others. **c** Ratio of phosphate (measured by determining total inorganic phosphorus) and protein amount present in the soluble fractions or in the membrane, determined by BCA assay. Data points correspond to three technical replicates. Error bars represent ±S.D.

efficient than the detergent reference but still solubilized approximately two-thirds of all MPs (Fig. 2a).

**Different xMALPs show distinct patterns in total MP solubilization.** One of the most attractive characteristics of the xMALPs is the possibility to directly solubilize MPs from the lipid bilayer into lipid nanodiscs: the extracted proteins are never removed from their native membrane environment. For this reason, xMALPs are often called native nanodiscs[12–15]. In light of the observed differences in the efficiencies of protein solubilization (see last paragraph), we next investigated the differences in protein content. Sodium dodecyl sulfate–polyacrylamide gel electrophoresis (SDS-PAGE) analysis allowed for a qualitative comparison of the protein content in the xMALPs and DDM micelles with the original membrane pellet (Fig. 2b). In order to prevent excessive smearing in SDS-PAGE, the polymer was largely removed from the proteins by chloroform/methanol protein precipitation[16]. Overall, most proteins that occur in the *E. coli* membrane sample are extracted into the lipid nanodiscs, except by SMA(2.3:1), which seems less efficient for higher molecular weight (MW) proteins. This trend extends to all polymers when solubilizing Jurkat membranes, where high MW proteins seem to be less efficiently extracted. In addition, some xMAs seem to preferentially solubilize certain proteins in comparison with other xMAs (see, for example, star-indicated bands at 75 kDa in the SMA-QA lane and at 18, 30, and 34 kDa in the SMA(3:1) lane of Fig. 2b). Surprisingly, all nanodiscs are unable to solubilize an *E. coli* protein appearing as an intense gel band in the 35 kDa region.

For a first qualitative evaluation of the lipid contents of the nanodiscs, we analyzed the ratio of phosphate and protein. This ratio gives an indication of the total amount of PLs per protein amount. In general, the nanodiscs have a similar phosphate/protein ratio compared with the native membrane (Fig. 2c). For *E. coli*, the nanodiscs show a slightly increased phosphate-to-protein ratio, which could indicate enrichment in PL in those

nanodiscs. The observed trend does not occur when solubilizing Jurkat membranes.

**The PL headgroup composition of the soluble *E. coli* lipidome depends on the xMA used.** In light of the observed differences in lipid and protein content among the nanodiscs, we proceeded to analyze in detail their lipid composition.

For the analysis of *E. coli* membrane-derived samples, we used shotgun lipidomics with a method able to analyze PL and glycerolipids (GLs). Unfortunately, the method used herein did not allow to measure cardiolipin (CL). Measuring PL, GL, and CL simultaneously is far from trivial[17,18]. Although some improved liquid chromatography (LC)-MS/MS methods started to emerge in order to bridge this gap, these are not yet broadly implemented.

The GL content was always <0.3%, being the highest in the untreated membrane and significantly lower in DDM and the nanodiscs. Remarkably, no GLs were detected in SMA-QA nanodiscs.

PLs are the main lipid class in *E. coli*, therefore we analyzed in detail their composition characteristics for each of the solubilized fractions (see Fig. 3a). The most abundant species was phosphatidylethanolamine (PE), being 86% of the total PLs present in the native membrane. This percentage was significantly reduced in all cases, most remarkably in SMA(2.3:1) (75%), DIBMA (71%), and SMA-QA (75%). The phosphatidylglycerol (PG) content was in all cases significantly increased (up to 22% PG content using SMA(2.3:1)) compared with the native membrane (12% PG). Other PLs, namely, phosphatidylinositol (PI), phosphatidylserine (PS), and phosphatidic acid (PA), were only identified in trace amounts. The DDM solubilization afforded significantly decreased amounts of those lipids (0.8%) compared to the membrane (1.2%). On the contrary, SMA-QA nanodiscs were enriched in PS, PA, and PI (2.5%).

Next, we investigated the fatty acid (FA) carbon chain length and saturation (Fig. 3b–d). Except when using DIBMA, for the

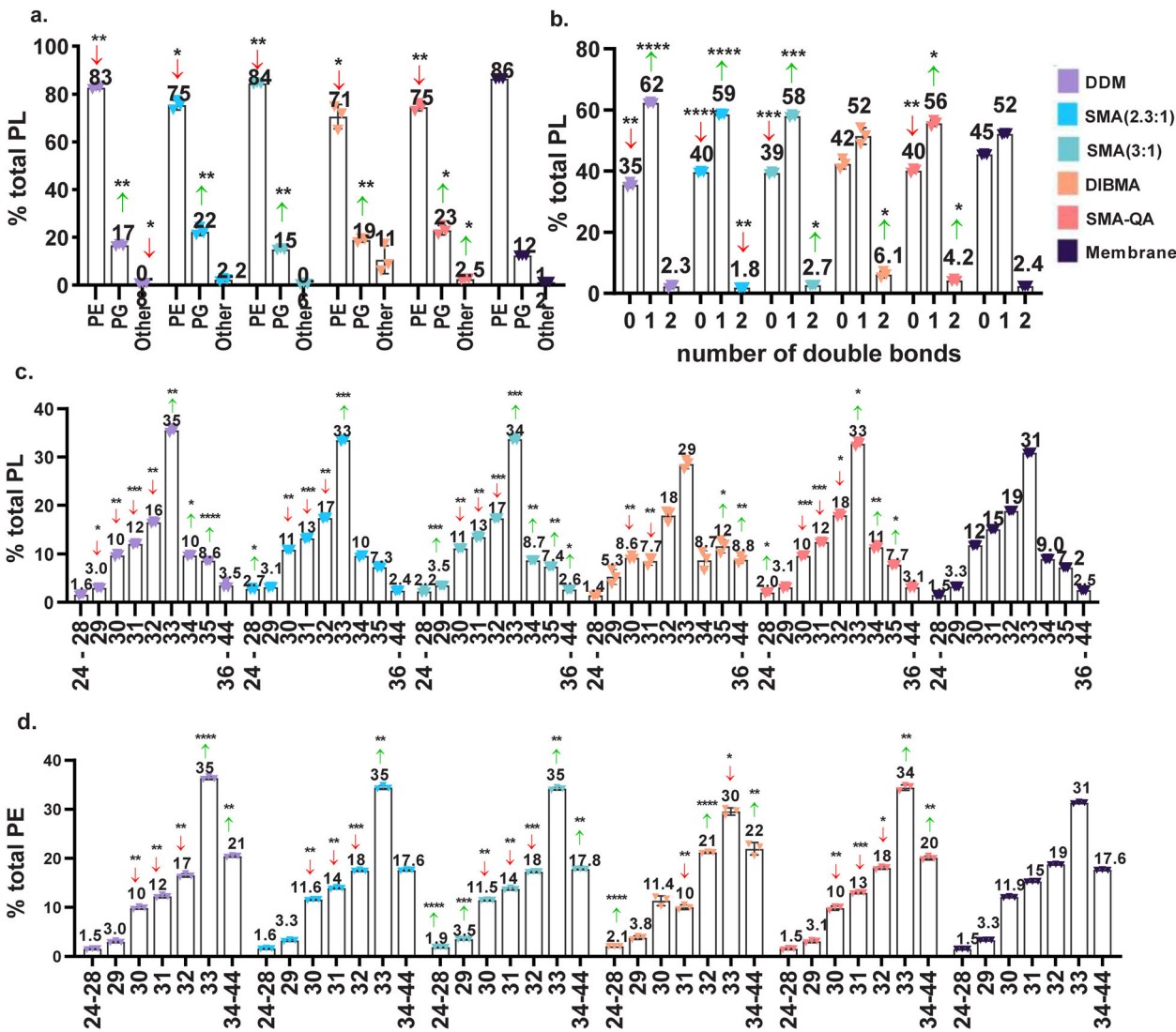

**Fig. 3 PL content in solubilized and unaltered membrane of *E. coli* analyzed by LC-MS/MS. a** *E. coli* headgroup distribution of PL species. The column "other" includes PS, PI and PA headgroups. **b** *E. coli* saturation of PL species. The number of double bonds indicated are those found in the total fatty acid chains of a single PL species. **c** *E. coli* distribution of carbon atoms present in fatty acid chains of the total PL species. **d** *E. coli* distribution of carbon atoms present in fatty acid chains of the total PE species. Data points correspond to three technical replicates. Error bars represent ±S.D. Significant differences (upon one-way ANOVA) are denoted as *$p < 0.05$, **$p < 0.01$, ***$p < 0.001$, ****$p < 0.001$.

rest of solubilizing methods the saturated FA were depleted compared with the membrane; concurrently the percentage of unsaturated FA was increased (Fig. 3b and ESI Fig. S1). In DDM and the nanodiscs formed by unmodified SMA, lipid species with one unsaturation were most increased compared with the membrane pellet. Interestingly, in DIBMA-derived nanodiscs, the lipid species containing two unsaturations underwent the largest increase and were 2.3 times more abundant than in the membrane pellet. In the case of SMA-QA, both 1 double bond (db) and 2 db species were increased in by 9 and 78%, respectively. Note that *E. coli* lacks polyunsaturated FAs, therefore the lipids containing two dbs comprise lipids with two monounsaturated FA acyl chains.

In order to get insight in the chain length distribution, we determined the total number of carbons present per PL species and compared the different profiles with the membrane pellet (Fig. 3c). For clarity, we refer to short (24–28 carbons), medium (29–32 carbons), long (33–34 carbons) and very long (from 36 carbons) lipid species. All solubilizing agents showed less medium

length species than the reference membrane. The content of long species was increased in DDM micelles. In the SMA(2.3:1) and SMA-QA, the short and long species were more abundant than in the membrane. In the case of DIBMA, the reduced presence of medium length species was compensated by an increased content of very long species (Fig. 3c, green and red arrows). The distinct distribution of FA chain lengths was also observed when analyzing specific PL headgroups (see Fig. 3d for PE and ESI Fig. S2 for PG). However, for some specific PL headgroups (e.g., PI), the chain length composition was virtually equal to the membrane (ESI Fig. S2).

**The composition of the soluble Jurkat lipidome varies for different xMALPs.** For the analysis of Jurkat membrane-derived samples, we used shotgun lipidomics with a method able to analyze PLs, GLs and sphingolipids (SLs). Unfortunately, cholesterol esters could not be reliably measured using this method. As an alternative, we measured cholesterol

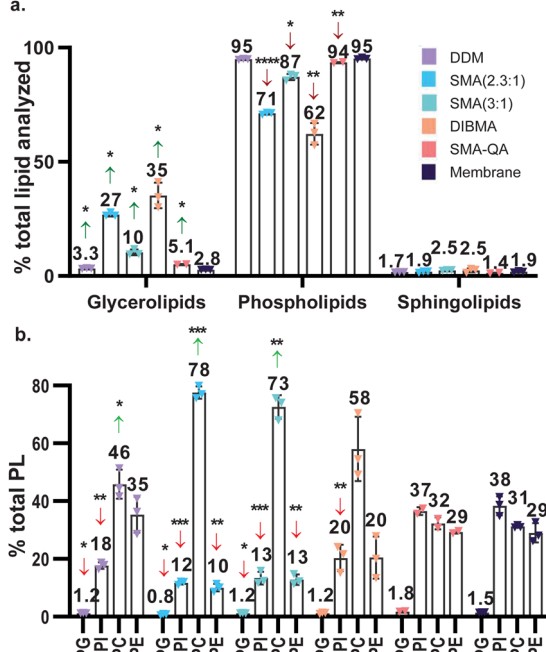

**Fig. 4 Lipid composition of solubilized membranes and membrane pellet of Jurkat cells analyzed by LC-MS/MS, which allowed for GL, PL, and SL identification. a** Jurkat cell lipid composition. Lipid class distribution in each solubilized sample compared with the membrane pellet. **b** Jurkat cells headgroup distribution of PL species. Data points correspond to three technical replicates, except for technical duplicates for SMA-QA. Error bars represent ±S.D. Significant differences (upon one-way ANOVA) are denoted as *$p < 0.05$, **$p < 0.01$, ***$p < 0.001$, ****$p < 0.001$.

using a separate fluorometric assay, the results of which are discussed later.

The membrane pellet was found to contain mainly PL (95%) and small amounts of GL (2.8%) and SL (1.9%) (Fig. 4a). The lipid composition of the DDM and the SMA-QA solubilized samples was practically identical to that of the native membrane, except for a minor increase in the GL content (3 and 5% of total lipids, respectively). In contrast, unmodified SMAs and DIBMA samples were depleted in PL, which was compensated by an increase in GL. This general trend was more pronounced when using DIBMA (62% PL, 35% GL), followed by SMA(2.3:1) (71% of PL, 27% GL) and SMA(3:1) (87% PL, 10% GL).

Different classes of SL were identified but the SL composition and content did not experience significant variations among the samples, except for the SMA-QA sample in which the content of lactosylceramides was reduced compared to the membrane (ESI Fig. S3).

In all the analyzed samples, we identified diacylglycerol (DG) and triacylglycerol (TG) species. Whereas in the membrane pellet TG:DG ratio was 1:3, in all solubilized samples the presence of TG was decreased in favor of DG. This ratio was approximately 1:7 when using DDM or SMA-QA and lowered to 1:32 for the unmodified SMAs or DIBMA (ESI Fig. S4).

The lipidomics analysis of PL showed that the membrane pellet contained 1.5% PG, 38% PI, 31% PC, and 29% PE. The same headgroup distribution was conserved in the lipidome solubilized by SMA-QA. Remarkably, the solubilization using unmodified SMAs showed a reduction of PG, PI, and PE (up to 1.8-, 3-, and 3-fold, respectively), in favor of PC that was 2.5-fold increased. This effect was more pronounced for SMA(2.3:1) than for SMA(3:1). A similar trend was observed when using DDM or DIBMA, but the differences compared with the membrane were less evident; for

example, the DDM sample displayed a 1.5-fold increase in PC content (Fig. 4b).

We observed that the detailed composition of the PC varied depending on the solubilization method used (ESI Fig. S5). In the membrane, the total PC fraction was composed of 88% intact PC and 12% PC of which the glycerol backbone had lost a FA chain (lysoPC). The use of DDM showed a threefold increase in lysoPC mostly at the cost of PC and, to a minor degree, of ether lipids (PC-O and PC-P). The detailed composition of PE remained more constant with the only change being the DIBMA PE-P content, which decreased to 10% compared to 21% found in the membrane (ESI Fig. S5).

**The FA saturation degree does not substantially affect xMALP solubilization of Jurkat membranes.** At this point, we inspected the degree of saturation in each sample. The global lipidome of the solubilized samples showed marked differences in the saturation degree, compared with the membrane (ESI Fig. S6). However, we hypothesized that the variation in the different lipid species between samples (see above) had an effect on total degree of saturation, provided that different lipid species display a distinct distribution of unsaturated FA. To test this idea, we proceeded to analyze the saturation degree of SL, GL, and each of the PL headgroups individually. Indeed, we observed that the changes in saturation profile compared with the membrane were very moderate for GLs and minimal for SLs and PL headgroups. In the GL fraction, the unmodified xMA preferentially solubilized saturated species. Interestingly, all SMAs were enriched in GL with an even number of dbs. Strikingly, in the DIBMA fraction the monounsaturated species represented 62% of the total GL (compared with 19% in the membrane pellet). This increment is mainly due to a high abundance of the DG (14:0/18:1) in the samples. The SL fraction remained virtually equal to the membrane for all xMA solubilizing conditions, whereas DDM showed enrichment of certain unsaturated species (ESI Fig. S6). For the PL headgroups, the most noticeable changes were found for PC-containing lipids in DDM and unmodified SMAs (ESI Fig. S7). Some minor changes were seen for PE-containing lipids solubilized with SMA(2.3:1), SMA-QA and DIBMA. Despite those, the xMAs mostly mimicked the membrane saturation degree for each PL headgroup (ESI Fig. S7).

**The FA chain length has little effect on xMA solubilization of Jurkat membranes.** The FA chain length influences the thickness of the lipid bilayer. For this reason, we compared the number of carbons—as an estimate for the FA length—of the solubilized lipid species with those in the membrane. Similar to the unsaturation trends (see above), FA length profiles of the total lipid species looked dramatically different for different samples. We again analyzed lipid classes (SL, GL, and the PL headgroups) separately in order test whether the differences are caused by the FA chain lengths themselves or by the distinct distribution of lipid species in the various xMA lipid nanodiscs.

For the FA length of SL, no major differences were observed between samples. As occurred for the unsaturations, the GL displayed most differences in chain length (Fig. 5c). Here two major aspects are worth highlighting: first, in all but the SMA-QA sample, there is a moderate to high reduction of lipids with 64 and more carbon atom chains. This observation matches with the reduction of TG species in favor of DG (see above). Accordingly, the presence of shorter chain species is increased. Second, the DIBMA-solubilized GL show a 11.5-fold increase of the 32 carbon species compared with the membrane. This is mainly due to the increased presence of the DG (14:0/18:1), previously highlighted as the cause of the increase of GL species

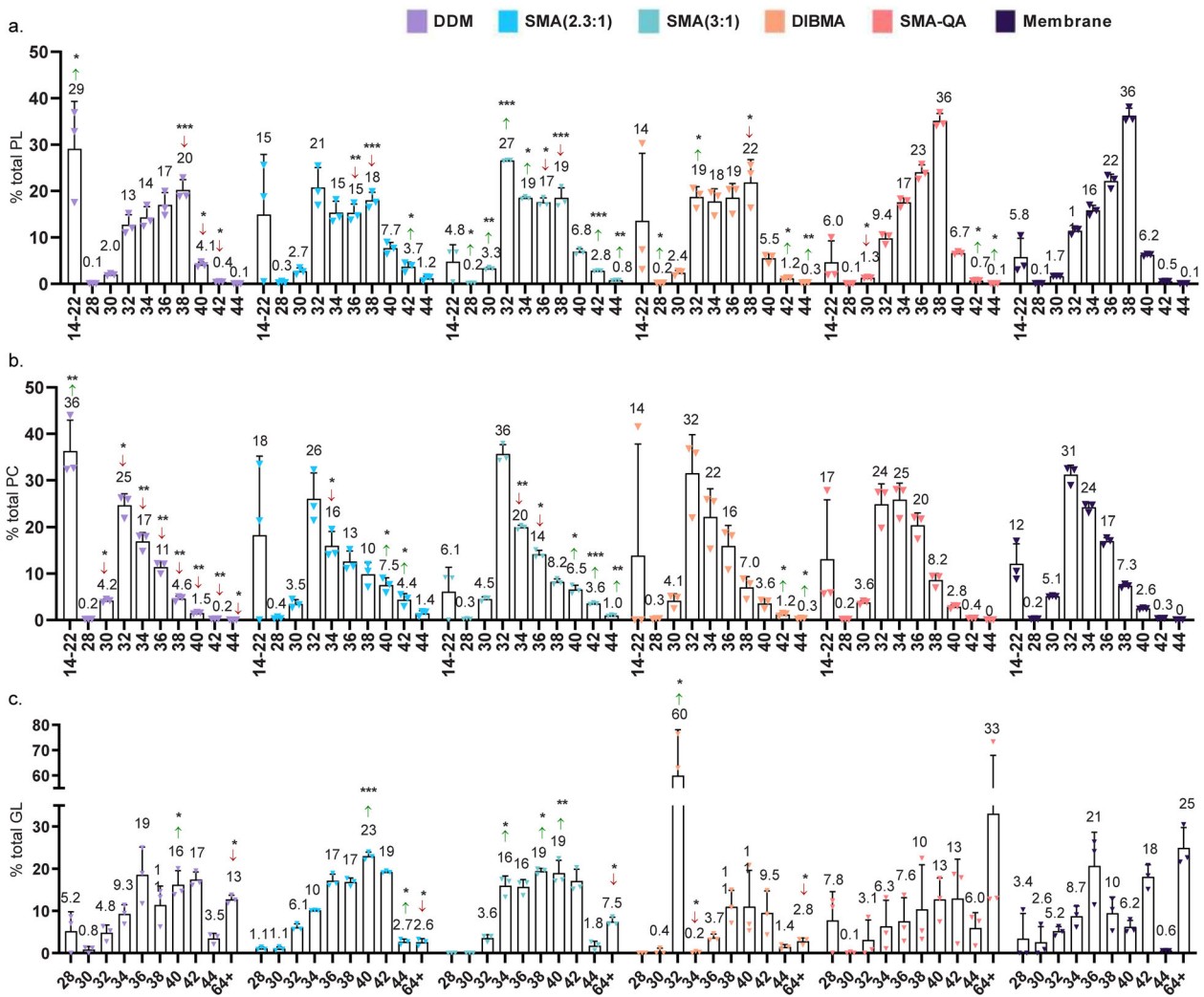

**Fig. 5 Fatty acid chain length of solubilized membranes and membrane pellet of Jurkat cells analyzed by LC-MS/MS.** Distribution of carbon atoms present in FA chains of **a** all PL species, **b** PC species, and **c** GL species from Jurkat solubilized membranes and membrane pellet. The PC length profiles observed for the solubilized samples largely resemble to that of the membrane, with some differences observed for the detergent sample that was enriched in PC species with 4–6 and ≥8 double bonds and depleted of 1–2 and 7 double bonds. Contrarily, the unmodified SMA samples were enriched in 6 and more double bond PC species and SMA(2.3:1) depleted of saturated and monounsaturated PC lipids. In general, the differences seen in total PL are therefore a result of the distinct headgroup composition of the samples. Data points correspond to three technical replicates, except for technical duplicates for SMA-QA. Error bars represent ±S.D. Significant differences (upon one-way ANOVA) are denoted as *$p < 0.05$, **$p < 0.01$, ***$p < 0.001$.

with 1 db (see above). The analysis of each PL headgroup species individually revealed that the carbon chain length profiles largely resemble that of the reference membrane (Fig. 5a and ESI Figs. S8 and S9). The most remarkable difference is found for the DDM-solubilized PC, which is 3-fold enriched in lysoPC (14–22 carbon species) with a consequential depletion in the rest of species (Fig. 5b).

**Cholesterol content in anionic xMAs is membrane like but substantially altered in DDM micelles and SMA-QA nanodiscs.** The cholesterol content was analyzed using the Amplex™ Red Cholesterol Assay Kit (see Supplementary Data Set 3). The method of determination differs from the lipidomics used for the rest of the mammalian lipids, therefore the cholesterol results are not presented as the percentage of total lipid content. The cholesterol content of unmodified SMAs and DIBMA resembled that in the membrane, being 42–46 nmol cholesterol/mg of protein versus 51 nmol cholesterol/mg of protein in the membrane. The solubilization with DDM was found to be 2.3 times enriched in

cholesterol. On the contrary, barely any cholesterol was found in the SMA-QA sample, the content of which was >90% reduced compared with the membrane sample (Fig. 6).

## Discussion

In this work, we compared the solubilization of native biological membranes by different membrane-solubilizing xMAs and DDM detergent. We showed that xMALP nanodiscs prepared with different xMA polymers display substantial quantitative and qualitative differences in their protein and lipid content. These differences occur in bacterial membranes (E. coli) as well as in mammalian membranes (Jurkat cell line).

Protein solubilization efficiency of xMAs is adequate. Our results show that the xMAs efficiently solubilize MPs from bacterial and mammalian membranes. In accordance with previous reports[16], we found that DDM is slightly more efficient for solubilizing E. coli MPs than any of the tested xMAs. Remarkably, SMA-QA consistently presented the lowest efficiency. A computational study by Xue et al. suggested that the membrane-

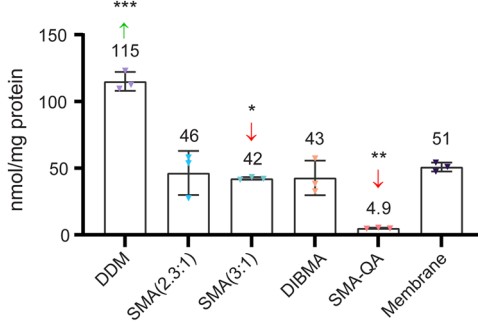

**Fig. 6 Cholesterol (nmol per mg of protein) in Jurkat membranes and solubilized membranes using the Amplex™ Red Cholesterol Assay Kit (triplicate measurements).** Error bars represent ±S.D. Significant differences compared with the membrane (upon individual *t* tests) are denoted as $*p < 0.05$, $**p < 0.01$, $***p < 0.001$.

disrupting properties of xMAs are linked to the stabilization of the xMALPs by interaction of the hydrophobic substituents with the bilayer interface while orienting their charged substituents to water-filled pores created by membrane defects[19]. The lower overall charge of SMA-QA compared with other xMAs (i.e., one positive charge instead of two negative charges per maleic acid unit) might therefore explain its reduced solubilization properties. It will be interesting for future research to address this matter by comparing solubilization efficiencies of SMA-QA with different ratios of styrene to maleimide quaternary amines, which will provide distinctly charged SMA-QA polymers. Despite its lower efficiency, the choice for SMA-QA over other xMAs may be justified because it can be used at acidic pH without precipitating, as is the case for other xMAs[9].

xMALP protein content varies for different xMAs. Our experiments show that xMAs display preferential solubilization of MPs in both prokaryotic and eukaryotic membranes. This behavior has been observed in the past when using SMA(3:1) and DIBMA with *E. coli* membranes[16,20,21]. Of the solubilizing agents tested, DDM affords the most homogeneous solubilized protein content, probably due to its complete disruption of the lipid bilayer. Of the xMAs, DIBMA and SMA(3:1) showed the largest resemblance to the protein content present in the membrane, although some specific proteins as well as higher MW proteins seemed missing (Fig. 2b). One may wonder what the cause of the observed differential solubilization could be. Although the inherent properties of a particular MP, such as total charge or number of transmembrane domains, may have an impact[22,23], we do not exclude that the distinct lipid preferences of different xMAs (see below) will also affect the solubilization of proteins in specific lipid microenvironments. Future research on proteins with known localization in lipid domains will be needed to further address this question.

**xMALPs show distinct distribution of lipid species.** To ascertain whether xMAs preferentially solubilize certain lipids from biological membranes, we analyzed the lipid composition of the differently solubilized membranes. The transfer of lipids and proteins between nanodiscs has been described for DIBMALPs[24] and SMALPs[25,26]. The effect in the nanodisc composition of this collisional lipid transfer under specific conditions of pH and ionic strength is far from negligible. However, the samples analyzed here are not purified and represent the total of the solubilized membrane. Therefore, the collisional lipid transfer among nanodiscs does not affect our readout. As we only detected PL species in the *E. coli* derived samples, we cannot comment on preferential solubilization of different lipid species from bacterial membranes. However, in the mammalian samples, we found

differences in the solubilized lipid species. The relative abundance of GL was markedly increased in SMA(3:1), SMA(2.3:1), and DIBMA. A possible reason may be the hydrolysis of the PL headgroup, leading to conversion of a PL species into a GL species. Chemically, this is unlikely in view of the high stability of the phosphodiester bond. Enzymatically this process may be possible, if xMAs would superiorly solubilize phospholipase C. Another, probably more likely, explanation may be the preferential solubilization of GL over PL by xMAs.

We did not observe considerable differences in the solubilization of SL and cholesterol for most of the xMAs. However, within the PL species, there were distinct preferences of xMAs for PG (increased in *E. coli* samples) and PC (increased in Jurkat samples), whereas PE was generally disfavored. We wondered what may be the driving force between these observed preferences and considered several molecular characteristics. The overall charge does not seem to play a role, because the solubilization behavior of PC and PE, both of which have a zwitterionic headgroup and a net zero charge, is generally opposite of each other. Being uncharged also does not provide an explanation, as the uncharged GL are increased, but cholesterol is not. Previous work with model membranes suggested that solubilization is not regulated by the properties of individual lipid constituents but by the overall physical state of the bilayer[27]. The lipid bilayer is structured in ordered (Lo) and disordered regions (Ld). Some lipid species, such as cholesterol and SL, are preferentially found in Lo regions. To date, there is no consensus on which phase is more prone to xMALP formation[27–30]. Nevertheless, it is known that Lo regions are more sensitive than Ld regions to perturbations induced by surfactant insertion into the lipid bilayer[31]. Hence, the fact that SL and cholesterol, found in Lo regions, are conserved in all unmodified SMA and DIBMA points at more accurate Lo solubilization, in line with this hypothesis. Other lipid species, such as PL and DG, can be found in both phases depending on specific lipid properties (see next paragraph). The effect of DG in the membrane packing is still controversial: they are reported by some to have a condensing effect, increasing the acyl chain order and bilayer thickness[32], but by others are seen as membrane order disruptors[33].

PL headgroup composition varies between xMALPs. The properties of PL largely depend on their headgroup. The headgroups not only confer distinct chemical properties such as charge and polarity but also influence the overall lipid shape. The size balance between the headgroups and the FA defines the PL geometry: PL with large headgroups such as PC or PG have a cylindrical geometry and tend to form flat membranes, whereas smaller headgroups give rise to a conical structure, leading to a curvature in the membrane[34]. The observation in our study that PG (in *E. coli* membranes) and PC (in mammalian membranes) have higher occurrence in the xMA lipid nanodiscs also supports the hypothesis that xMA polymers may disrupt highly structured regions of the membrane more easily. Specifically, an increase in the occurrence of these lipids in the xMA nanodiscs reflects that lipid species in more disordered regions are less well extracted and solubilized by xMAs.

Preferential lipid solubilization is linked to xMA chemical properties. The comparative lipidomics analysis in this study shows that the specific xMA determines the composition of the soluble lipidome. Our observations are in contradiction with the extended idea that xMAs are promiscuous solubilizing agents that do not preferentially extract lipids from the membrane. This discrepancy could be explained by several factors. First of all, only a few studies have analyzed the lipid content of SMALPs derived from biological membranes[13,15,21,35–37]. In these reports, the utilized SMA was SMA(2:1), which is less hydrophobic than the SMAs we used here. The different chemical properties of our SMAs may account for the discrepancy with the results of

the studies which conclude that there is no preferential lipid extraction by SMA(2:1), even though some point out the preferential solubilization of unsaturated FA chains by SMA (2:1)[36]. Second, a couple of studies using SMA(3:1) reported non-preferential lipid extraction, but these have mainly used analysis by thin-layer chromatography (TLC) and quantification by densitomety[38]. The LC-MS analysis used here is more sensitive and may reveal differences less well detectable by TLC, especially when these changes are subtle. Third, SMA(3:1) showed clear preferential lipid solubilization in Jurkat membranes (see Fig. 4), but when applied to *E. coli* membranes, there were only minor differences with the composition of the membrane. This latter result is in line with previous studies that used LC-MS to study the lipid composition of SMA(3:1) *E. coli* nanodiscs[21,37]. The majority of studies toward lipid solubilization by xMAs, however, have used model membranes consisting of only two or three lipid species and lacking protein content[27–29]. In the absence of proteins, lipids behave differently than in biological membranes (e.g., distinct phase partitioning)[34,39], suggesting that the xMA lipid preference in native membranes may not occur in model lipid bilayers. We here provide data which suggest that the overall membrane properties rather than the individual lipid species regulate xMA solubilization, underlining the importance of using biological membranes as the study object.

In conclusion, we have analyzed the composition of xMALPs derived from eukaryotic and prokaryotic membranes using LC-MS/MS and SDS-PAGE. The lipid and protein content varied between xMALPs and compared with the native membrane. The solubilization efficiency of the xMAs versus a widely used detergent was only slightly decreased for prokaryotic membranes and equally efficient for eukaryotic membranes. These results show that the differences in solubilization efficiency between xMAs and detergent do not prevent the choice for the copolymers. In addition, the small decrease in the solubilization efficiency is counterbalanced by the reported increased protein stability that the xMAs offer in exchange[5,7].

The distinct protein solubilization observed here indicates that certain MPs may benefit from the use of a specific xMA. More detailed proteomics-based comparative studies will be necessary to answer the question why xMAs preferentially solubilize particular MPs and which factors govern these preferences. Until that question is resolved, it will be beneficial to test various xMAs when solubilizing and purifying a specific MP of interest. For now, our analysis reveals that xMAs have preferential protein and lipid solubilization characteristics that vary with the chemical structure of the applied xMA.

Here we also provided evidence that the solubilization preferences are guided by the bulk of lipid species but not by the FA composition (i.e., length and degree of saturation). This suggests that the shape of the lipid species or, more general, the degree of order in the structure of the membrane could be the determining factor for solubilization, whereas the membrane thickness or packing strength has a much milder effect. To get more insight into the detailed membrane features that control xMALP formation, complex models that mimic biological membranes will be necessary. Altogether, this may lead to the future design of novel polymers with improved membrane-disruption properties or to chemically tuning them to extract specific membrane regions.

The preferential lipid extraction reported here has important implications for future analysis of xMALP lipid contents and reveals the importance of the correct controls. For example, the use of xMALPs in the analysis of lipids that immediately surround MPs (annular lipid shell) may suffer from a bias stemming from the inherent preferences of the xMA. This can be addressed by analyzing the lipid content of unpurified xMA nanodiscs as control. Another recommendation involves the

analysis of the insoluble fraction after xMA solubilization: this will reveal whether the xMALPs with a MP of interest are representative for the total protein or only a small fraction.

Our initial question was to what extent xMA nanodiscs resemble the native membrane. This study revealed that the resemblance depends on the used xMA. Ideally, a more membrane-like environment would be preferred. From our results, we conclude that SMA-QA is the most membrane like if looking only at lipid content and disregarding cholesterol. Nevertheless, when both protein and lipid content are taken into account, SMA(3:1) gives the most balanced solubilization.

Finally, this study has also shown that the differences in protein and lipid content in xMALPs are more subtle in *E. coli* than in Jurkat membranes. This make us wonder how membranes from other species such as yeast or extremophiles would behave with different xMAs. Future studies detailing the xMA solubilization of such systems would help sketching a complete picture of xMA solubilization preferences. Some of our future efforts are along these lines and will be reported in due course.

## Methods
XIRAN® (SL25010 S25; SL30010 S30; SZ25010) and Sokalan C9 were a kind gift from Polyscope and BASF, respectively: XIRAN® SL25010 S25 (3:1 styrene-to-maleic acid ratio; MW 10,000 g/mol); XIRAN® SL30010 S30 (2.3:1 styrene-to-maleic acid ratio; MW 6500 g/mol), XIRAN® SZ25010 S25 (3:1 styrene-to-maleic anhydride ratio; MW 10,000 g/mol); and Sokalan C9 (diisobutylene-to-maleic acid ratio 1:1 and MW 15,300 g/mol). All other materials were purchased from commercial vendors and used without prior purification.

**Statistics and reproducibility**. Statistical analysis was performed using Prism 8 (GraphPad Software, USA), including multiple unpaired *t* tests with Welch correction and one-way Brown–Forsythe–Welch analysis of variance with Dunett's correction. The number of independent biological experiments for each panel is highlighted in the figure legends.

**Lipid nomenclature**. The lipid nomenclature and classification follow the recommendations of the International Lipid Classification and Nomenclature Committee (LIPID MAPS)[40,41].

**Preparation of SMA and DIBMA**. In all, 37% HCl (∼7 mL) was slowly added to 20 mL of Sokalan C9 solution, XIRAN® SL25010 S25, or XIRAN® SL30010 S30, until acidic pH was reached (pH < 2, checked by pH paper). The precipitated white solid was washed with 30 mL of miliQ water and spun down at $4000 \times g$ for 30 min. The supernatant was discarded, and the same wash procedure repeated three more times. The wet white solid was snap-frozen in liquid nitrogen and lyophilized to dry powder. From the lyophilized xMA powder, 6% (w/v) stock solutions in 50 mM HEPES and 0.5 M NaCl were created. To this end, the pH was adjusted to 8 by dropwise addition of NaOH (10 M at first, later 1 M) while sonicating in an ultrasound water bath to foster the complete dissolution of the xMA. The stock solutions were stored at −20 °C in the dark.

**Preparation of styrene maleimide quaternary ammonium (SMA-QA)**. SMA-QA was prepared following the reported procedure of Ravula et al.[9]. In short, 770 mg of SMA anhydride (XIRAN® SZ25010 S25; 3:1 styrene-to-maleic anhydride ratio; MW 10,000 g/mol) and 1 g of 2-aminoethyltrimethylamonium chloride (5.71 mmol) were dissolved in 23 mL of anhydrous dimethylformamide. To the solution, 3.84 mL of triethylamine (26.89 mmol) were added and the whole was heated to 100 °C under stirring for 20 h. Once the solution was at room temperature, the product was precipitated using diethyl ether. This precipitate was recovered upon vacuum filtration and washed three times with diethyl ether. The dried intermediate was added to 30 mL acetic anhydride containing 660 mg of sodium acetate and 200 mg of triethylamine and the whole was heated to 100 °C overnight. Once the solution was at room temperature, the product was precipitated using diethyl ether and washed three times with this same solvent. The crude product was then dissolved in water and passed through a Sephadex G10 column. The product was lyophilized and characterized using infrared (IR). In order to prepare the working SMA-QA stocks, 60 mg of pure product were dissolved in 1 mL of 50 mM HEPES and 0.5 M NaCl buffer.

**Production and solubilization of *E. coli* membranes**. *E. coli* strain DH5α were grown in 500 mL LB media at 37 °C for 16 h. The cells were pelleted at 3000 rpm at 4 °C for 30 min. Pelleted cells were lysed by French Press in lysis buffer (20 mM HEPES, pH 7.4; 0.1 M NaCl; 10% glycerol; complemented with Roche Complete inhibitor cocktail). The mixture was centrifuged $3000 \times g$, 15 min at 4 °C to remove

unbroken cells and at $100,000 \times g$, 60 min, 4 °C to pellet the membrane fraction. The membrane pellet was resuspended in buffer 50 mM HEPES-NaOH, pH 7.8, and 0.3 M NaCl to a concentration that will afford 30 mg membrane pellet per mL of buffer after addition of solubilizing agent. The resuspended pellet was solubilized by addition of 2.5% wt/v of solubilizing agent from a previously prepared 6% stock solution. The solubilizing agents used were: styrene–maleic acid copolymer (SMA(3:1) or SMA (2.3:1), respectively), SMA-QA, DIBMA, or DDM. The mixture was left rocking for 2 h at 37 °C when using xMA or at 4 °C when using DDM. Centrifugation at $100,000 \times g$, 40 min, 4 °C removed unsolubilized membranes. The supernatants containing the solubilized membrane were separated from the insoluble pellet and used in further experiments.

### Preparation of solubilized Jurkat cell membranes (adapted from refs. [42,43]).
Cells ($3.4 \times 10^8$) were resuspended in 18 mL of ice-cold hypotonic buffer (10 mM HEPES pH 7.5, 42 mM KCl, 5 mM $MgCl_2$, protease inhibitor mixture), incubated on ice for 15 min. The suspension was then passed 10× through a 26-gauge needle. The supernatant was divided into 18 aliquots of 1 mL and centrifuged for 20 min at $25,000 \times g$, 2 °C to pellet the membranes. Three aliquots were kept separate and each of the remaining 16 aliquots were resuspended to a concentration of 20 mg/mL in a buffer containing 50 mM HEPES pH = 7.8, 0.3 NaCl, benzonase, and 2% solubilizing agent (DDM, SMA(2.3:1), SMA(3:1), DIBMA, or SMA-QA). The samples were rocked overnight at 37 °C if using xMA or at 4 °C if using DDM. After this time, the samples were centrifuged for 45 min at $100,000 \times g$, 4 °C. The supernatants containing the solubilized membrane were separated from the insoluble pellet and used in further experiments.

### Phosphate and protein concentration determination.
Protein concentrations were determined using the Pierce™ BCA Protein Assay Kit (Thermo Scientific™), according to the instructions by the vendor.

For determination of the amount of PLs in xMALPs, samples (triplicates; 50 μL) and different amounts of a phosphate standard (0.65 mM; Sigma-Aldrich P3869) were lyophilized to remove solvent. Next, samples were treated with 90 μL of 8.9 N $H_2SO_4$ and heated at 230 °C for 25 min. After cooling down, 30 μL of hydrogen peroxide was added and heating was continued for 30 min or until the samples were fully transparent. Samples were diluted with 330 μL water. Next, 50 μL of 2.5% ammonium molybdate in water (w/v) and 50 μL of 10% ascorbic acid in water (w/v) were added. The mixture was heated to 100 °C for 10 min and absorption at 820 nm was read using a Molecular Devices ID3 plate reader. Standard curves ($R^2 = 0.99$) and unknowns were calculated using Microsoft Excel.

### Protein precipitation[16].
In order to prevent smearing of protein bands in SDS-PAGE caused by excess polymer, the proteins present in the solubilized fractions were precipitated to separate them from the polymer. An aliquot of ice-cold sample was mixed with four volumes of ice-cold $CH_3OH$ by vortexing. Next, one volume of ice-cold $CHCl_3$ was added. The sample was mixed by vortexing and then three volumes of ice-cold $H_2O$ were added and mixed the same way. After centrifugation ($13,000 \times g$ for 5 min at 4 °C), the top aqueous layer was removed carefully not to disrupt the protein layer formed at the interface. An additional four volumes of ice-cold $CH_3OH$ were added and mixed by vortexing, and the sample was pelleted by centrifuging for 15 min at $15,000 \times g$ at 4 °C. Finally, the $CH_3OH$ was removed and the samples were snap-frozen and lyophilized to eliminate possible traces of solvent. Once dry, the protein pellets were resuspended in $H_2O$ and the protein concentration was determined using a BCA Assay Kit (standard curve $R^2 = 0.99$). SDS buffer was added to a sample volume containing 15 μg of protein and subjected to SDS-PAGE.

### Fourier-transform infrared (FTIR) spectroscopy.
FTIR spectra were recorded on a Bruker Vertex 70 spectrometer. Direct examination of the products was done by attenuated total reflectance (ATR) using the Bruker ATR platinum set-up. The recorded spectra were analyzed by the OPUS software Absorption values are expressed as wavenumbers ($cm^{-1}$); only relevant absorption bands are given. For xMA IR spectra, see ESI—Supplementary Method.

### Shotgun lipidomics (E. coli)—see Supplementary Data Set 1.
The E. coli shotgun lipidomics was performed by Lipotype GmbH (Dresden, Germany). The Lipotype Shotgun Lipidomics platform consists of automated extraction of samples, an automated direct sample infusion, and high-resolution Orbitrap MS including lipid class-specific internal standards to assure absolute quantification of lipids. An in-house developed software—LipotypeXplorer—is used for identification of lipids in the mass spectra.

*Lipid extraction.* Lipids were extracted using chloroform and methanol[44]. Samples were spiked with lipid class-specific internal standards prior to extraction. After drying and resuspending in MS acquisition mixture, lipid extracts were subjected to MS analysis.

*Spectra acquisition.* Mass spectra were acquired on a hybrid quadrupole/Orbitrap MS equipped with an automated nano-flow electrospray ion source in both positive and negative ion mode.

*Data processing and normalization.* Lipid identification using LipotypeXplorer[45] was performed on unprocessed (*.raw format) mass spectra. For MS-only mode, lipid identification was based on the molecular masses of the intact molecules. MS/MS mode included the collision-induced fragmentation of lipid molecules and lipid identification was based on both the intact masses and the masses of the fragments. Prior to normalization and further statistical analysis, lipid identifications were filtered according to mass accuracy, occupation threshold, noise, and background. Lists of identified lipids and their intensities were stored in a database optimized for the particular structure inherent to lipidomic datasets. Intensity of lipid class-specific internal standards was used for lipid quantification.

### Shotgun lipidomics (Jurkat cell line)—see Supplementary Data Set 2
*Lipid extraction.* In all, 700 μL of sample (diluted in water and containing a total of 10 μg of protein) was mixed with 800 μL 1 N $HCl:CH_3OH$ 1:8 (v/v), 900 μL $CHCl_3$, and 200 μg/mL of the antioxidant 2,6-di-tert-butyl-4-methylphenol (Sigma Aldrich). In all, 3 μL of SPLASH® LIPIDOMIX® Mass Spec Standard (#330707, Avanti Polar Lipids) was spiked into the extract mix. The organic fraction was evaporated using a Savant Speedvac spd111v (Thermo Fisher Scientific) at room temperature, and the remaining lipid pellet was stored at −20 °C under argon.

*Mass spectrometry.* Just before MS analysis, lipid pellets were reconstituted in 100% ethanol. Lipid species were analyzed by LC–electrospray ionization–MS/MS on a Nexera X2 UHPLC system (Shimadzu) coupled with hybrid triple quadrupole/linear ion trap mass spectrometer (6500+ QTRAP system; AB SCIEX). Chromatographic separation was performed on a XBridge amide column (150 mm × 4.6 mm, 3.5 μm; Waters) maintained at 35 °C using mobile phase A [1 mM ammonium acetate in water–acetonitrile 5:95 (v/v)] and mobile phase B [1 mM ammonium acetate in water–acetonitrile 50:50 (v/v)] in the following gradient: (0–6 min: 0% B → 6% B; 6–10 min: 6% B → 25% B; 10–11 min: 25% B → 98% B; 11–13 min: 98% B → 100% B; 13–19 min: 100% B; 19–24 min: 0% B) at a flow rate of 0.7 mL/min, which was increased to 1.5 mL/min from 13 min onwards. SM, CE, CER, DCER, HCER, and LCER were measured in positive ion mode with a precursor scan of 184.1, 369.4, 264.4, 266.4, 264.4, and 264.4, respectively. TAG, DAG, and MAG were measured in positive ion mode with a neutral loss scan for one of the fatty acyl moieties. PC, LPC, PE, LPE, PG, LPG, PI, LPI, PS, and LPS were measured in negative ion mode by fatty acyl fragment ions. Lipid quantification was performed by scheduled multiple reactions monitoring (MRM), the transitions being based on the neutral losses or the typical product ions as described above. The instrument parameters were as follows: Curtain Gas = 35 psi; Collision Gas = 8 a.u. (medium); IonSpray Voltage = 5500 and −4500 V; Temperature = 550 °C; Ion Source Gas 1 = 50 psi; Ion Source Gas 2 = 60 psi; Declustering Potential = 60 and −80 V; Entrance Potential = 10 and −10 V; Collision Cell Exit Potential = 15 and −15 V. The following fatty acyl moieties were taken into account for the lipidomic analysis: 14:0, 14:1, 16:0, 16:1, 16:2, 18:0, 18:1, 18:2, 18:3, 20:0, 20:1, 20:2, 20:3, 20:4, 20:5, 22:0, 22:1, 22:2, 22:4, 22:5 and 22:6, except for TAGs which considered: 16:0, 16:1, 18:0, 18:1, 18:2, 18:3, 20:3, 20:4, 20:5, 22:2, 22:3, 22:4, 22:5, 22:6.

*Data analysis.* Peak integration was performed with the MultiQuant™ software version 3.0.3. Lipid species signals were corrected for isotopic contributions (calculated with Python Molmass 2019.1.1) and were normalized to internal standard signals. Unpaired T test p values and false discovery rate corrected p values (using the Benjamini/Hochberg procedure) were calculated in Python StatsModels version 0.10.1.

**Reporting summary.** Further information on research design is available in the Nature Research Reporting Summary linked to this article.

## Data availability
All data generated or analyzed during this study are included in this published article, its supplementary information, and supplementary data files. Any remaining information can be obtained from the corresponding author upon reasonable request.

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

## Acknowledgements
We acknowledge Polyscope and BASF for kindly providing the SMA and DIBMA polymers. We thank Jef Callebaut for helping with scripts for data analysis, Suravi Chakrabarty for help in cell culture, and Jonas Dehairs for collecting lipidomics data of Jurkat samples. We acknowledge funding from Horizon2020 - Marie Curie fellowship (no. 752252) to M.B.-X., FWO for a post-doctoral fellowship (no. 12Y0720N) to M.B.-X., the Ministerium für Kultur und Wissenschaft des Landes Nordrhein-Westfalen, the Regierende Bürgermeister von Berlin–inkl. Wissenschaft und Forschung, and the Bundesministerium für Bildung und Forschung.

## Author contributions
M.B.-X. carried out experiments and analyzed the data. M.B.-X. and S.H.L.V. conceived the project, interpreted data, and wrote the paper.

## Competing interests
The authors declare no competing interests.
