## [Peer Review File · Communications Biology]

Reviewers' comments:

Reviewer #1 (Remarks to the Author):

In this study, Barniol-Xicotá and Verhelst investigate the usefulness of different maleic acid-based copolymers (xMAs) to prepare membrane proteins within native-like lipid environments. To that end, the authors applied a broad range of known xMAs, as well as the detergent DDM as a control, to solubilise membrane proteins from both prokaryotic (E.coli DH5 α strain) and eukaryotic (Jurkat cells) membranes. They focus their analysis on resultant protein content, protein-lipid ratio, and the lipid compositions obtained within the xMA nanoparticles. All experiments have been conducted in a comprehensible way and state of the art technology (LC-ESI/MS/MS) has been used to determine the lipid composition with respect to headgroup distribution, saturation, and chain length.

These maleic acid-based co-polymer systems (xMA) are increasingly being used to study membrane proteins within a more native-like lipid environment. The manuscript is therefore timely, acts as a useful resource for the xMA community, and would be welcomed by the membrane protein researchers. The crucial question of this study was to what extent different xMA nanodiscs resemble the lipid composition of the native membrane. The authors showed that the obtained lipid compositions varied depending on which xMA was chosen as the solubilisation agent. The experiments are robustly performed, and the figures are of suitable quality. The statistical analysis is appropriate and valid.

The main concern is the take-home message delivered in the title and the abstract. The authors express "preferential" lipid solubilization by xMAs which is difficult to assess from their dataset – indeed the discussion around their data is considered, which is not reflected in the title. Additionally, the authors take the line that "some of these nanodiscs are less native than initially thought". This line is contentious, and the underlying point is not true, as many researchers in the field do not believe this and this opinion should not be presumed. Instead, those that use xMA co-polymers understand that they offer the best solution to study membrane proteins within lipid compositions encountered in the cellular milieu, avoiding the need for reconstitution using potentially destabilising detergents, but are unlikely to give you the exact cellular context. Indeed, the manuscripts main strength may even be that it shows that xMA's consistently provide a lipid composition akin to the cellular membrane compositions, with nuanced changes found, supporting the idea which they refute in the abstract.

To rectify this, it is recommended that the manuscript title and abstract be refocused onto its strengths, which is a "Lipidomic and in-gel assessment of maleic acid co-polymer-based nanodiscs". This will help to avoid misinterpretation of their study. Additionally, a broader discussion is requested on how their data could be interpreted in consideration to the points provided below:

- As we know from previous studies, the composition of the surrounding lipid annular belt of membrane proteins is likely different compared to bulk lipids. Therefore, do the observed lipid compositions originate from the different membrane proteins being solubilised instead of solubilisation preference by different xMA? Especially as the authors report a different protein population being solubilised for each xMA type. A control experiment using the different nanodiscs to solubilise a target protein could bring more lucidity.
- The authors do not discuss the discovery that different xMA nanodiscs can undergo collisional transfer and lipid exchange in solution – which can lead to 'averaging' of the lipid composition. Please see for examples: <https://doi.org/10.1016/j.ymeth.2020.04.010>, <https://doi.org/10.1021/acs.langmuir.6b02927> and <https://doi.org/10.1038/srep45875>. These studies have changed the way the field thinks about xMA nanodiscs, the new dogma being that a native lipid-

bilayer core is likely preserved which represents a composition akin to those found within its cellular host (and not its direct lipid environment). The nuanced changes observed could be due to collisional transfer and not "preferential solubilisation".

- Each different xMA co-polymer can accommodate a different disc size and, therefore, a different amount of lipids. Could this lead to the differences found in relative lipid composition between different xMA's? Moreover, are the differences in disc size also related to the difference in protein content observed, i.e. could larger MP's or MP complexes with larger transmembrane regions be solubilised by xMA's with larger discs? This is currently hard to assess as no disc size measurements are provided/reported, e.g. by dynamic light scattering, which would enable the author and reader to assess their data better.
- Could the author comment on whether the buffer conditions could have a significant impact on the parity between membrane and xMALPS, e.g. salt concentrations, pH, etc? Have they performed any experiments to this effect? Especially as increasing the Mg²⁺ and Ca²⁺ content aids in DIBMA solubilisation (see: DOI: 10.1016/j.chemphyslip.2019.03.004).
- The authors consistently find substantial amount of PC lipids in their E. coli experiments. PC lipids are not found within E. coli (see: <https://doi.org/10.1093/femsre/fuv008>). Can the authors explain why they observe >6% PC within their E. coli membrane samples? Were these contaminants on the LC? If so, if you eradicated this contribution to the total PL% then do you end up with significantly different lipid compositions between xMA's and the membrane fraction?
- It is seen that SMA-QA is the worst solubilising xMA. However, this polymer stock is prepared in a very different way to the other xMA. Do the authors envisage that the polymer preparation method could directly impact its solubilisation activity?
- It is advised that the authors provided meta data and tables (maybe in an Excel format) for lipidomic data, so that other researchers can access and use this data.

Minor revisions (see below) would also substantially help for further clarification:

- Page 3 Figure 1: m for DIBMA is not illustrated in the picture. Either add m to the pictograph or remove it in the caption.
- Page 6 Figure 3: Why does the bargraph indicate a significant decrease in C30 lipids for SMA copolymers but the percentage is with 12% the same as in the membrane pellet.
- Page 8: Text does not refer to figure 4A.
- Page 10: "For the PL headgroups, the most noticeable changes were found for PC containing lipids in DDM and unmodified SMAs (Fig. 5B)." – Figure 5B does not represent this statement.
- Page 11: It would help to refer to the different bargraphs individually rather than just once to figure 5.
- Page 13: Text does not refer to figure 6.
- Page 23: I assume that the applied flow rate was in $\mu\text{L}/\text{min}$ rather than in mL/min

Reviewer #2 (Remarks to the Author):

Recently detergent-free extraction of membrane proteins using polymer, attracting more attention due to its application in the membrane protein field. In the current manuscript, the authors study the effect of different polymers and their impact on the membrane solubilization and its selectivity in the lipid composition of extracted lipids. The authors used bacterial and mammalian cell lines to study different polymer solubilization efficiencies. Primarily authors use various techniques to analyze the lipid content and found that distinct variation in their lipids based on the polymer chemical properties used for the solubilization. Overall this is a well-executed manuscript and written clearly to explain the observed results. I would recommend this manuscript should be published after a minor revision.

What is the approximate molecular weight of each polymer used? This information should be included in the manuscript (figure 1).

Authors used the same weight ratio of the polymer in all the studies, Authors should comment on what will be the effect of molecular weight on the preferential lipid solubilization.

Is there any effect of polymer to lipid ratio on the solubilization efficiency?

SDS page gel looks faint and needs improvement if possible.

Figure 3B- Change the x-axis label as the "number of double bonds"

The decreased efficiency of SMA-QA is attributed to low overall charge of the polymer. Being the low molecular weight of SMA-QA(assuming the authors used the same starting material as a reference from the Ravula et al.), showed no difference in the Lipid compositions in the case of mammalian membranes but with less efficiency. This effect can also be due to the less polymer available to solubilize the membrane. The author should perform the experiment with a different ratio and see if this is true?

Figure 2C: Why the standard deviation of SMA (2.3:1) is high compared to other samples?

Point-by-point discussion

We thank the reviewers for their critical assessment of the manuscript and for their insightful comments. We have addressed the remarks raised and we provide a point by point description of the changes made in the manuscript.

Reviewer 1:

The main concern is the take-home message delivered in the title and the abstract. The authors express "preferential" lipid solubilization by xMAs which is difficult to assess from their dataset – indeed the discussion around their data is considered, which is not reflected in the title. Additionally, the authors take the line that "some of these nanodiscs are less native than initially thought". This line is contentious, and the underlying point is not true, as many researchers in the field do not believe this and this opinion should not be presumed.

We agree that the quoted sentence could be misleading and confuse the reader, and we decided to delete this sentence. We have also adjusted the title of the manuscript (see next point)

Instead, those that use xMA co-polymers understand that they offer the best solution to study membrane proteins within lipid compositions encountered in the cellular milieu, avoiding the need for reconstitution using potentially destabilising detergents, but are unlikely to give you the exact cellular context. Indeed, the manuscripts main strength may even be that it shows that xMA's consistently provide a lipid composition akin to the cellular membrane compositions, with nuanced changes found, supporting the idea which they refute in the abstract. To rectify this, it is recommended that the manuscript title and abstract be refocused onto its strengths, which is a "Lipidomic and in-gel assessment of maleic acid co-polymer-based nanodiscs". This will help to avoid misinterpretation of their study.

We agree with the reviewer that the xMA co-polymers offer the best solution to date to study membrane proteins. In fact, we state this idea in our manuscript as well, mentioning that the xMAs are a better solubilization method than detergents e.g. in the sentences: "These results show that the differences in solubilization efficiency between xMAs and detergent do not prevent the choice for the co-polymers. In addition, the small decrease in the solubilization efficiency is counterbalanced by the reported increased protein stability that the xMAs offer in exchange". We also agree with the reviewer that the composition of the xMAs is akin to the membrane, as we show in the manuscript. Nevertheless, it is also true that there are changes in the lipid composition depending on the xMA used for solubilization. Although those differences in lipid content do not suppose a dramatic change compared to the native membrane, they are remarkable and significant enough to be mentioned. We also believe that stating the differences in lipid

and protein solubilization between the xMAs, may help researchers when choosing for a xMA for their specific experiments. To avoid possible misinterpretations of our findings, we now changed the title to “Lipidomic and in-gel analysis of maleic acid co-polymer-based nanodiscs reveals differences in composition of solubilized membranes”

Additionally, a broader discussion is requested on how their data could be interpreted in consideration to the points provided below:

• As we know from previous studies, the composition of the surrounding lipid annular belt of membrane proteins is likely different compared to bulk lipids. Therefore, do the observed lipid compositions originate from the different membrane proteins being solubilised instead of solubilisation preference by different xMA? Especially as the authors report a different protein population being solubilised for each xMA type. A control experiment using the different nanodiscs to solubilise a target protein could bring more lucidity.

This work aimed to clarify if there would be differences in the protein solubilization and lipid content of the xMA nanodiscs, depending on the chemical nature of the xMA used. We agree that defining the reason or origin of the observed differences in the xMALPs content is, definitely, a matter of interest. Nevertheless, this is a far from trivial question to address and we feel this is out of the scope of the present work.

The reviewer asks if the differences in lipid content observed are just an effect of “preferential” protein solubilization by the xMAs. Although at the moment we cannot prove or refute this hypothesis, we need to take several factors into account.

(1) First of all, the proposed mechanism of xMALP formation by Xue et al. ([10.1016/j.bpj.2018.06.018](https://doi.org/10.1016/j.bpj.2018.06.018)) and Orekhov et. al ([10.1021/acs.langmuir.8b03978](https://doi.org/10.1021/acs.langmuir.8b03978)), it seems unlikely that the proteins rather than the lipids will drive the solubilization and, therefore the differences observed. As shown by Xue et al, the SMA co-polymers bind and adsorb to the lipid bilayer intercalating their hydrophobic groups in the lipids and later using the hydrophilic charged groups to interact with water molecules and open a pore which will with time lead to the nanodisc formation. Hence, according to their results and that the lipids are the most abundant species in the bilayer, the solubilization is not likely to be driven (exclusively) by the proteins present in the bilayer and not be affected by the nature of the lipids. Therefore, although we agree that the annular lipids of the different solubilized proteins may also contribute, we doubt that those account for the total of the lipid differences observed.

(0) secondly, it must be taken into account that we do not purify the lipid nanodiscs with particular embedded proteins – i.e. our resulting sample is a mixture of all nanodiscs with proteins, but likely also without proteins. Although we agree that it would be interesting to analyze the lipid content of annular belts around particular proteins, it

would be necessary to analyze various target proteins using different xMAs, which is not the purpose of the current work. In addition, as the xMALPs undergo collisional lipid transfer at different rates depending on the nature of the xMA used (e.g. SMA3:1 transfer efficiency is 40 times higher than SMA2.3:1 under the same conditions) as proven by Grethen et al [10.1007/s00232-018-0024-0](https://doi.org/10.1007/s00232-018-0024-0) , when analyzing the lipid content of purified nanodiscs the results may be not be comparable among polymers. Therefore, one may need to correct for differences in lipid transfer rate (e.g. adjusting ionic strength) for each case. On top of that, analyzing a single protein may not be representative and one might want to compare solubilization efficiencies of several target proteins, analyze the lipid content of the xMAs and then counterpose it with our current data. As explained above, the experiments needed to elucidate the question posed by the reviewer, embodies a project on its own and we may consider carrying this out in future investigations.

• *The authors do not discuss the discovery that different xMA nanodiscs can undergo collisional transfer and lipid exchange in solution – which can lead to ‘averaging’ of the lipid composition. Please see for examples: <https://doi.org/10.1016/j.ymeth.2020.04.010>, <https://doi.org/10.1021/acs.langmuir.6b02927> and <https://doi.org/10.1038/srep45875>. These studies have changed the way the field thinks about xMA nanodiscs, the new dogma being that a native lipid-bilayer core is likely preserved which represents a composition akin to those found within its cellular host (and not its direct lipid environment). The nuanced changes observed could be due to collisional transfer and not “preferential solubilisation”.*

We have not discussed the collisional lipid transfer that takes place among nanodiscs as this effect is not relevant in our case. We work with unpurified xMALPs and because we analyze the total of the solubilized lipidome, the lipid interchange between nanodiscs does not affect our readout. Therefore, we can be sure that the changes observed are not an artifact from collisional lipid transfer. This wouldn't be the case, of course, if we would be analyzing single protein nanodiscs as in that case the lipid composition in the xMALP would be subjected to this effect and therefore the possible “preferences” in lipid solubilization, could be masked.

In order to clarify this point, we added an explanation and the suggested references in the main text. Now it reads “(...)The transfer of lipids and proteins between nanodiscs has been described for DIBMALPs¹ and SMALPs^{2,3}. The effect on the nanodisc composition

of this collisional lipid transfer under specific conditions of pH and ionic strength is far from negligible. However, the samples analyzed here are not purified and represent the total of the solubilized membrane. Therefore, the collisional lipid transfer among nanodiscs does not affect our readout.(...)”

- *Each different xMA co-polymer can accommodate a different disc size and, therefore, a different amount of lipids. Could this lead to the differences found in relative lipid composition between different xMA's? Moreover, are the differences in disc size also related to the difference in protein content observed, i.e. could larger MP's or MP complexes with larger transmembrane regions be solubilised by xMA's with larger discs? This is currently hard to assess as no disc size measurements are provided/reported, e.g. by dynamic light scattering, which would enable the author and reader to assess their data better.*

During the work, we have done some preliminary assessment of disc size by DLS and in all cases the samples are polydisperse with disc sizes ranging from approx. 10 to 100 nm. This again, may be a result of working with an unpurified sample and we find hard to make a statement that this accounts for the differences in lipid composition. It may well be that the differences in disc size distribution have an effect on the differences in protein content observed among the different nanodiscs. In the future it will be interesting to analyze in depth the protein content of different xMALPs. This will allow to assess if there's a specific class of protein, e.g. complexes, that is better solubilized with a certain xMA and draw specific relations between xMA properties and protein solubilization.

- Could the author comment on whether the buffer conditions could have a significant impact on the parity between membrane and xMALPS, e.g. salt concentrations, pH, etc? Have they performed any experiments to this effect? Especially as increasing the Mg²⁺ and Ca²⁺ content aids in DIBMA solubilisation (see: DOI: 10.1016/j.chemphyslip.2019.03.004).

We have not experimentally assessed what the effect of different buffer conditions are in the parity between membrane and xMALPs, as in this work we wanted to compare different xMALPs under the same buffer conditions. However, it has been show in other papers ([10.1002/anie.201610778](https://doi.org/10.1002/anie.201610778) or [10.1038/nprot.2016.070](https://doi.org/10.1038/nprot.2016.070) or [10.1016/j.bbamem.2019.183125](https://doi.org/10.1016/j.bbamem.2019.183125)) that the buffer conditions do affect xMALP formation. For example, although DIBMALP formation can be accelerated by the addition of divalent cations, the SMALPs do not tolerate divalent cations. In the same way high pH > 9 favor DIBMALP formation but are detrimental for SMALP formation. In order to avoid buffer effects, we used the same buffer for all the xMALPs, which did not contain any

detrimental elements for the efficiency any of the polymers used (e.g. divalent cations for SMA). We picked "standard" buffer conditions that are tolerated by all polymers and used in several previous papers. We have now indicated this explicitly in the main text (beginning of the results section) by adding the following sentence: "The polymers used for solubilization display different compatibilities with buffering systems. For example, SMA-QA is compatible with low pH⁴ whereas solubilization by DIBMA is stimulated by lower pH⁵ (*insert ref). To prevent any favorable or unfavorable effects of the buffer composition, we chose for buffer conditions that are tolerated by all polymers. Specifically, these comprised 50 mM HEPES pH 8.0 and 0.5 M NaCl"

- The authors consistently find substantial amount of PC lipids in their E. coli experiments. PC lipids are not found within E. coli (see: <https://doi.org/10.1093/femsre/fuv008>). Can the authors explain why they observe >6% PC within their E. coli membrane samples? Were these contaminants on the LC? If so, if you eradicated this contribution to the total PL% then do you end up with significantly different lipid compositions between xMA's and the membrane fraction?

We are thankful to the reviewer to spot this oversight of us. We now investigated the origin of this "PC" species, which indeed, turned out to be an artifact. In order to correct this, we have re-analyzed the E. coli lipid data eliminating the contribution of the PC species and made new graphs in the cases that was necessary (Figure 3, Figure S1, Figure S2). Despite these modifications in the graphs, the overall trends observed have not substantially changed.

- *It is seen that SMA-QA is the worst solubilising xMA. However, this polymer stock is prepared in a very different way to the other xMA. Do the authors envisage that the polymer preparation method could directly impact its solubilisation activity?*

We do not have a reason to believe that the polymer preparation method would impact its solubilization activity. In fact, the polymer stock is not prepared substantially different than the other stocks. The main difference is that SMA-QA is not submitted to acid washes, which main purpose is to precipitate the polymers to be able to wash away possible impurities. SMA-QA does not undergo this treatment because a) SMA-QA is synthesized in house and b) SMA-QA is acid resistant, so it does not precipitate under treatment with acid. It is therefore purified through sephadex columns, as reported in the original paper in Angew Chem. As this treatment also washes away possible impurities, we are sure that the polymer is pure, as confirmed also via IR. For the rest, equally to all xMAs used, SMA-QA is lyophilized to dryness and the solid material is then

redissolved in the same buffer (from the same stock) as all the other xMAs and pH adjusted.

- *It is advised that the authors provided meta data and tables (maybe in an Excel format) for lipidomic data, so that other researchers can access and use this data.*

We will make all the lipidomics data available in Excel format as supplemental information on the journal's webpage.

Minor revisions (see below) would also substantially help for further clarification:

- *Page 3 Figure 1: m for DIBMA is not illustrated in the picture. Either add m to the pictograph or remove it in the caption.*

We changed the figure and now the "m" is added.

- *Page 6 Figure 3: Why does the bargraph indicate a significant decrease in C30 lipids for SMA copolymers but the percentage is with 12% the same as in the membrane pellet.*

This is because, for space reasons in the figure, we decided to show rounded up values. In the membrane the C30 are 12.1%, in the SMAs they are 11.6% and 11.5%. All of those, however, are rounded to 12%. When analyzing the data, this difference is shown to be statistically significant so, we indicated it. We agree that this is confusing. We have now changed it providing the first decimal value in the figure itself in this specific case but also in all other cases where something similar occurred.

- *Page 8: Text does not refer to figure 4A.*

We have now fixed this.

- *Page 10: "For the PL headgroups, the most noticeable changes were found for PC containing lipids in DDM and unmodified SMAs (Fig. 5B)." - Figure 5B does not represent this statement.*

We have corrected the text, which now refers the right figure S7 instead.

- *Page 11: It would help to refer to the different bargraphs individually rather than just once to figure 5.*

We made changes in the text to link each sentence with the corresponding bargraph and now reads:

"As occurred for the saturations, the GLs displayed most differences in chain length (Fig. 5C)." "The most remarkable difference is found for the DDM solubilized PC, which is 3-fold enriched in lysoPC (14-22 carbon species) with a consequential depletion in the rest of species (Fig. 5B)." "The analysis of each PL headgroup species individually revealed that the carbon chain length profiles largely resemble that of the reference membrane (Fig. 5A and ESI Fig. S8-9)."

- *Page 13: Text does not refer to figure 6.*

Now this is fixed

- *Page 23: I assume that the applied flow rate was in $\mu\text{L}/\text{min}$ rather than in mL/min*
The 0.7 mL per minute (or = 700 μL per minute) is correct. The column used has an internal diameter of 4.6 mm, for which a typical flow rate is 1 mL/minute..

Reviewer #2

We thank the reviewer for the nice comments, and we proceed to address the minor revisions the reviewer suggested.

What is the approximate molecular weight of each polymer used? This information should be included in the manuscript (figure 1).

We have now included this information in the Materials and Methods section and reads: "XIRAN® (SL25010 S25; SL30010 S30; SZ25010) and Sokalan C9 were a kind gift from Polyscope and BASF respectively. XIRAN® SL25010 S25 (3:1 styrene to maleic acid ratio; Mw 10000 g/mol); XIRAN® SL30010 S30 (2.3:1 styrene to maleic acid ratio; Mw 6500 g/mol) , XIRAN® SZ25010 S25 (3:1 styrene to maleic anhydride ratio; Mw 10000 g/mol); and Sokalan C9 (diisobutylene to maleic acid ratio 1:1 and Mw 15300 g/mol). All other materials were purchased from commercial vendors and used without prior purification."

Authors used the same weight ratio of the polymer in all the studies, Authors should comment on what will be the effect of molecular weight on the preferential lipid solubilization. Is there any effect of polymer to lipid ratio on the solubilization efficiency?

Polymer to lipid ratio has been shown to be an important parameter for solubilization efficiency and disc size of the xMALP ([10.1016/j.bbamem.2019.183125](https://doi.org/10.1016/j.bbamem.2019.183125) , [10.1021/acs.langmuir.7b03742](https://doi.org/10.1021/acs.langmuir.7b03742)). In our study, we compare equal amounts in weight of the different polymers with respect to the membranes. We decided to keep the concentration equal as that is the parameter that is usually specified when describing xMA solubilization protocols. This is a common practice for these types of polymers. In addition, for DIBMA it has been shown that fractions of different molar masses were similarly efficient in solubilizing a saturated lipid⁵. As a 2% w/v concentration has been reported for various xMAs to give good solubilization efficiencies, we have decided for this particular weight ratio of the polymer.

We now mention this explicitly in the beginning of our results section with the following sentence: "We chose for a 2% (w/v) total xMA concentration during solubilization, as this concentration has shown efficient solubilization using different xMAs⁶.

SDS page gel looks faint and needs improvement if possible.

Unfortunately, the gel picture can not really be improved. We could have adjusted the contrast artificially with photoshop, but we preferred not to manipulate the image.

Figure 3B- Change the x-axis label as the "number of double bonds"

We have changed "n of double bonds" for "number of double bonds" as suggested by the reviewer.

The decreased efficiency of SMA-QA is attributed to low overall charge of the polymer. Being the low molecular weight of SMA-QA (assuming the authors used the same starting material as a reference from the Ravula et al.), showed no difference in the Lipid compositions in the case of mammalian membranes but with less efficiency. This effect can also be due to the less polymer available to solubilize the membrane. The author should perform the experiment with a different ratio and see if this is true?

We used a different starting material than Ravula and co-workers. Whereas they used poly(Styrene-co- Maleic Anhydride "1.3:1 with a molecular weight of "1600 g/mol, we have used a styrene maleic anhydride copolymer which has a ratio of styrene to maleic anhydride of 3:1 and a molecular weight of 10,000 g/mol. Our starting material is commercialized by Polyscope under the name of XIRAN® SZ25010. This co-polymer has the same molecular weight as SMA 3:1 here used and does not differ substantially with the mw of the other polymers in this study. Hence, the molecular weight of our in-house synthesized SMA-QA is comparable to that of the other SMA polymers we used in this work and has an overall higher charge per polymer particle than the low molecular weight SMA-QA. We have now indicated this more clearly in the results section by adding the sentence: "This polymer was synthesized in-house from a 10,000 g/mol SMA(3:1) anhydride (see methods section for details)", and in the methods section: "770 mg of styrene maleic acid anhydride (XIRAN® SZ25010 S25; 3:1 styrene to maleic anhydride ratio; Mw 10000 g/mol) and 1 g of 2-aminoethyltrimethylammonium chloride (5.71 mmol) were dissolved in 23 ml of anhydrous DMF... etc"

Obviously, SMA-QA still has less charges than the negatively charged SMA and DIBMA, which have two negative charges per MA instead of one positive charge per MA for SMA-QA, but this is inherent to this type of polymer. We hope this addresses the concerns of the reviewer. We have now speculated on the solubilization effect by the lower charge of SMA-QA in our discussion section in a sentence that reads as follows: "The lower overall charge of SMA-QA compared with other xMAs (i.e. one positive charge instead of two negative charges per maleic acid unit), might therefore explain its reduced solubilization properties.". We believe that this clears the concerns of the reviewer.

Figure 2C: Why the standard deviation of SMA (2.3:1) is high compared to other samples?

We performed our analysis in triplicates, for which one of those triplicates using SMA 2.3:1; the values obtained were very different from the other 2. To make sure that the observed variability was not an artifact of the analysis, we repeated the lipidomics analysis of that sample. Nevertheless, the results of the second lipidomics analysis are consistent with those found during the first sample analysis. Apparently, the replicates of this solubilization had higher variability than the other samples.

1. Danielczak, B. & Keller, S. Collisional lipid exchange among DIBMA-encapsulated nanodiscs (DIBMALPs). *Eur. Polym. J.* **109**, 206–213 (2018).
2. Hazell, G. *et al.* Evidence of Lipid Exchange in Styrene Maleic Acid Lipid Particle (SMALP) Nanodisc Systems. *Langmuir* **32**, 11845–11853 (2016).
3. Cuevas Arenas, R. *et al.* Fast Collisional Lipid Transfer among Polymer-Bounded Nanodiscs. *Sci. Rep.* **7**, 1–8 (2017).
4. Ravula, T., Hardin, N. Z., Ramadugu, S. K., Cox, S. J. & Ramamoorthy, A. Formation of pH-Resistant Monodispersed Polymer–Lipid Nanodiscs. *Angew. Chemie - Int. Ed.* **57**, 1342–1345 (2018).
5. Oluwole, A. O. *et al.* Formation of Lipid-Bilayer Nanodiscs by Diisobutylene/Maleic Acid (DIBMA) Copolymer. *Langmuir* **33**, 14378–14388 (2017).
6. Lee, S. C. *et al.* A method for detergent-free isolation of membrane proteins in their local lipid environment. *Nat. Protoc.* **11**, 1149–1162 (2016).

REVIEWERS' COMMENTS:

Reviewer #1 (Remarks to the Author):

Barniol-Xicotá, M. and Verhelst, S.H.L., have sufficiently revised the manuscript and their rebuttal was convincing when addressing the individual points made in the initial review. The change and choice of title better describes the work and I thank the reviewers for their open-mindedness to this request. They have excluded the erroneous PC data from the E. coli lipid data set and included the requested lipidomics metadata (which will greatly aid the community). The increased detail in polymer preparation, experimental details, comparisons to literature, and explanation of the data have made the manuscript substantially more robust. Overall, I believe this manuscript to now be entirely significant and suitable for publication in Communications Biology.

Reviewer #3 (Remarks to the Author):

All my queries have been answered by the authors. I recommend this article for publication.